# Hypermetabolism and Substrate Utilization Rates in Pheochromocytoma and Functional Paraganglioma

**DOI:** 10.3390/biomedicines10081980

**Published:** 2022-08-16

**Authors:** Ondřej Petrák, Denisa Haluzíková, Judita Klímová, Matěj Zítek, Robert Holaj, David Michalský, Květoslav Novák, Radka Petráková-Doležalová, Jan Kvasnička, Thi Minh Phuong Nikrýnová Nguyen, Zuzana Krátká, Martin Matoulek, Jiří Widimský, Tomáš Zelinka

**Affiliations:** 1Third Department of Medicine, First Faculty of Medicine and General Faculty Hospital, Charles University, 12800 Prague, Czech Republic; 2Institute of Sport Medicine, First Faculty of Medicine and General Faculty Hospital, Charles University, 12800 Prague, Czech Republic; 3First Department of Surgery, First Faculty of Medicine and General Faculty Hospital, Charles University, 12800 Prague, Czech Republic; 4Department of Urology, First Faculty of Medicine and General Faculty Hospital, Charles University, 12800 Prague, Czech Republic

**Keywords:** pheochromocytoma, functional paraganglioma, catecholamines, metanephrines, indirect calorimetry, resting energy expenditure, respiratory quotient, substrate metabolism

## Abstract

The overproduction of catecholamines in pheochromocytoma/paraganglioma (PPGL) induces a hypermetabolic state. The aim of this study was to evaluate the incidence of a hypermetabolic state and differences in substrate metabolism in consecutive PPGL patients divided by catecholamine phenotype. Resting energy expenditure (REE) and respiratory quotient (RQ) were measured in 108 consecutive PPGL patients and 70 controls by indirect calorimetry. Hypermetabolic state was defined according to the Mifflin St. Jeor Equation as a ratio above 110%. Hypermetabolic state was confirmed in 70% of PPGL patients, regardless of phenotype. Older age, prevalence of diabetes mellitus and arterial hypertension were correlated with hypermetabolic PPGL as compared to normometabolic form. Analysis according to overproduced catecholamine showed differences in VCO_2_ (*p* < 0.05) and RQ (*p* < 0.01) and thus different substate metabolism between phenotypes in hypermetabolic form of PPGL. Lipid utilization was higher in the adrenergic phenotype (*p* = 0.001) and positively associated with the percentage of REE ratio (R = 0.48, *p* < 0.001), whereas the noradrenergic phenotype preferentially oxidizes carbohydrates (P = 0.001) and is correlated with the percentage of REE ratio (R = 0.60, *p* < 0.001). Hypermetabolic state in PPGL is a common finding in both catecholamine phenotypes. Hypermetabolic PPGL patients are older and suffer more from diabetes mellitus and arterial hypertension. Under basal conditions, the noradrenergic type preferentially metabolizes carbohydrates, whereas the adrenergic phenotype preferentially metabolizes lipids.

## 1. Introduction

Pheochromocytoma and functional paraganglioma (PPGL) are catecholamine-producing tumors arising from the adrenal medulla or neural crest progenitors located outside of the adrenal gland, respectively [1,2]. Excessive production of catecholamines by the tumor often leads to many laboratory changes and clinical disturbances, including hypermetabolic state, accompanied by weight loss despite normal appetite and food intake [1,3]. In addition, significant weight gain occurs after successful tumor removal [3,4].

Catecholamines exert their metabolic effects through mobilization of fuels from their storage sites for oxidation in order to meet increased energy requirements [5]. According to previously published studies, adrenaline is more potent in lipolysis than noradrenaline [6,7]. The effect of long-term exposure to high plasma catecholamine concentrations on the utilization of each substrate has not yet been reported. Therefore, we decided to analyze the effect of catecholamines on basal metabolism. As our previous work with biochemical phenotypes shows, the biological action of noradrenaline is different in isolated overproduction (noradrenergic phenotype) and in combined overproduction with adrenaline (adrenergic phenotype) [8]. These two phenotypes differ significantly not only in terms of metabolic and hemodynamic properties but also in tumor morphology and genetic background [8,9,10,11].

Indirect calorimetry is the gold-standard technique for determining resting energy requirements. This method allows the volume of gas exchange (CO_2_ production and O_2_ consumption) to be measured, the proportion of oxidized nutrients to be determined and the energy released from each nutrient to be calculated. Hypermetabolism, defined as a significant increase of more than 10% in measured resting energy expenditure relative to predicted resting energy expenditure, is a multifactorial response involving adverse changes in several metabolic pathways (glycogenolysis, lipolysis and proteolysis) in many organs (liver, adipose tissue and muscle) [12]. Chronic hypermetabolism results in significant organ, muscle, protein and lipid wasting; hepatic steatosis; and immunosuppression [12]. Furthermore, hypermetabolism is a possible manifestation of increased biological activity of the tumor and may also be related to cardiovascular complications of PPGL [13].

The aim of our prospective study was to assess the incidence of hypermetabolism in a consecutive and unselected group of patients with PPGL and to evaluate the differences in substrate metabolism. These are calculated from calorimetry parameters and 24 h urine urea nitrogen excretion in PPGL patients with respect to the catecholamine biochemical phenotype in an attempt to assess the possibly differing effect of catecholamines on basal metabolism. We hypothesized that both catecholamine phenotypes lead to a hypermetabolic state with an increase in resting energy expenditure but that the presence of adrenaline is accompanied by a more pronounced basal lipid oxidation rate, expressed as a decrease in the respiratory quotient (RQ), in comparison with the noradrenergic phenotype. 

## 2. Materials and Methods

The study was designed as a prospective project. We consecutively included 108 patients with newly diagnosed PPGL (57 women), 92 patients with pheochromocytoma and 16 patients with paraganglioma during the period from January 2015 to February 2022. The diagnosis of PPGL was based on elevated plasma metanephrine and normetanephrine levels, as well as tumor demonstration by CT scan and/or PET/CT with fluorodeoxyglucose (FDG) or fluorodopa. Catecholamine biochemical phenotype was determined according to levels of plasma metanephrines [8,9]; to confirm the biological action of catecholamines, we determined either plasma (*n* = 58) or urinary catecholamines (*n* = 50). Thus, the phenotype was determined not only on the basis of metanephrines but also its own biological action. To better assess the biological effects of individual catecholamines, groups of patients producing only adrenaline, only noradrenaline and both were created. However, the group producing pure adrenaline did not differ from the group with combined overproduction in terms of basic and calorimetric parameters; therefore, these groups were merged into one group consisting of the adrenergic phenotype (Appendix A). Diagnosis of PPGL was definitively confirmed histopathologically after the surgery. All patients were examined during a short hospitalization in our department. Diabetes mellitus was defined as either medication with insulin/oral antidiabetic drugs, repeated fasting plasma glucose >7.0 mmol/L or glycated hemoglobin ≥ 48 mmol/mol. Germline gene mutation was detected in 10 patients (9%): 1 *TMEM*, 4 *NF-1*, 1 *SDHx*, 1 *MEN*, 2 *VHL* and 1 *MAX* mutation. Metastatic PPGL was confirmed in 3 patients. 

The control group consisted of 70 apparently healthy individuals, including 20 hospital staff, 30 volunteers and 20 patients before elective cholecystectomy in a stable phase of the disease (this group was part of our previous recent study [14]). No signs of inflammation or any severe chronic disease were present. Their basic laboratory tests (liver and renal functions, as well as electrolytes and fasting glucose levels) were normal.

Informed consent was obtained from all subjects. The ethical committee of our institution approved the study (approval date: 21 May 2015, code 20/15). The study was conducted in accordance with the Declaration of Helsinki.

Biochemical analyses, including metanephrine levels, were described in detail in our previous study [15]. Plasma catecholamines were extracted by commercial kit (Chromsystems, Munich, Germany). Analytes were analyzed by liquid chromatography with electrochemical detection (HPLC-ECD). Urine catecholamines were quantified by liquid chromatography with fluorescent detection (HPLC-FLD 1100S, Agilent 1100, Agilent Technologies, Inc., Wilmington, DE, USA). The system was calibrated with a catecholamine standard using a ClinRep test kit (Recipe Chemicals and Instruments GmbH, Munich, Germany). Additionally, 24 h urine samples were collected from all subjects with PPGL for photometric determination of urea nitrogen levels on a Modular analyzer (Roche Diagnostics, Paris, France). 

### 2.1. Energy Metabolism and Body Fat Mass Measurement

Resting energy expenditure (REE) and respiratory quotient (RQ) were analyzed with a computerized, open-circuit, indirect calorimetry system that measured resting oxygen uptake and resting carbon dioxide production using a ventilated canopy (Vmax Encore 29 N, VIASYS Healthcare Inc.; SensorMedics, Yorba Linda, CA, USA). Body fat percentage was measured by a Bodystat 1500 instrument (Bodystat Ltd., Isle of Man, UK) simultaneously with indirect calorimetry. Details were published in our previous article [3]. The tests were performed in the same room, in a quiet environment with constant temperature (23–25 °C). Individuals were placed in the supine position for at least 15 min before starting the test. Oxygen consumption (VO_2_) and carbon dioxide production (VCO_2_) were continuously evaluated for approximately 20 min, with data recorded every 5 s. The first 5 min were disregarded to ensure adequate acclimatization, and the mean of the last 15 min was considered in the analysis. The predicted REE value was calculated using the Mifflin–St. Jeor formula, which can provide a better estimate than the Harris–Benedict Equations derived in 1919, based on the subject’s gender, height, weight and age [16]. Hypermetabolic state was defined as an increase in a ratio between measured and estimated REE (≥110%), normometabolic state (91 to 109%) and hypometabolic state (≤90%). Due to the low incidence of hypometabolism in the PPGL group (2 patients), they were merged with the normometabolic group. All calorimetry measurements, including anthropometric, were carried out by one investigator to reduce intraoperator variability. RQ is an indicator of the carbohydrate-to-fat oxidation ratio. The RQ of carbohydrates is 1, the RQ of fat is 0.7 and the RQ of protein is approximately 0.81. Detailed analysis of substrate oxidation requires the measurement of urinary urea excretion for the assessment of protein oxidation and calculation of the non-protein RQ. Basal substrate metabolism (i.e., the protein, carbohydrate and the fat oxidation rates), was calculated from collected data according to Westenskow formulas [17].

### 2.2. Statistical Analysis

Statistical analysis was performed with Statistica for Windows version 12.0 (StatSoft, Inc., Tulsa, OK, USA). Data were described as means ± SD or medians (and interquartile range) for non-normally distributed variables, as assessed by the Shapiro–Wilks W test. Non-normally distributed variables (fasting plasma glucose, plasma catecholamine and metanephrine levels and urine catecholamines) were log-transformed (log_10_) before analysis. Two independent groups were tested by Student’s *t*-test or Mann–Whitney test, as appropriate. Differences among the groups according to overproduced catecholamine were analyzed by one-way analysis of variance (ANOVA), followed by a post hoc analysis. Categorical variables are expressed as frequencies (%) and were tested by chi-square or Fisher’s exact test. Pearson’s or Spearman’s correlation analysis was performed to characterize the relationship between individual variables. *p* < 0.05 was considered statistically significant.

## 3. Results

Table 1 shows the basic characteristics of the studied groups—PPGL and control group. The main significant differences were in fasting glucose levels and glycated hemoglobin, plasma metanephrine and normetanephrine levels (*p* < 0.001) and current smokers (*p* < 0.05). The prevalence of type 2 diabetes mellitus in the PPGL group was 29%, and that of persistent arterial hypertension was 68%. Alpha-blocker treatment was administered in 93% of PPGL patients, with a median dose of doxazosin of 3 mg daily, and beta-blocker was administered in 32% of the PPGL group. A proportion of 27% of patients with PPGL were treated with a statin, which was significantly higher than in the control group (*p* < 0.01).

Table 2 presents the results of indirect calorimetry between PPGL and control groups. There is a significant increase in VO_2_, VCO_2_ and measured REE in the PPGL group in comparison to the control group (*p* < 0.001). The percentage of REE ratio increased by 116% in the PPGL group vs. 101% in controls. According to the definition, hypermetabolism was present in 70% of PPGL subjects and in 16% of controls, whereas hypometabolism was observed in 2% and 8%, respectively. No change between groups was observed in RQ. 

Table 3 shows a comparison of basic clinical characteristics between hyper- and normometabolic PPGL subjects. Hypermetabolic PPGL patients are older (*p* < 0.05), with borderline prevalence of diabetes mellitus (*p* = 0.05) and parameters of glucose metabolism (*p* < 0.05 for glycated hemoglobin and *p* < 0.01 for fasting glucose levels), as well as an increased prevalence of arterial hypertension (*p* < 0.05). No differences were observed with respect to therapy, including a dose of doxazosin, or in multiples of catecholamine and metanephrine levels above the upper reference range. 

Table 4 consists of the results of indirect calorimetry parameters, with expected differences in VCO_2_, VO_2_ and REE, even after adjusting for body weight, body surface area and free fatty mass (*p* < 0.001). A more detailed analysis of substrate metabolism showed that whole hypermetabolic PPGL patients have a higher basal rate of lipid utilization compared to normometabolic patients (626 kcal/day vs. 891 kcal/day; *p* < 0.01).

Further analysis was focused on the hypermetabolic state of PPGL (Table 5). As expected, comparison of both catecholamine phenotypes with hypermetabolism showed that patients with an adrenergic phenotype are older (*p* < 0.01), with higher metanephrine levels (*p* < 0.001). Neither plasma normetanephrine values nor multiples of their plasma levels above the upper reference range differed between phenotypes. TSH levels were significantly decreased in noradrenergic compared to adrenergic phenotypes, although still within the normal reference range (*p* < 0.01), and another differences were detected in total cholesterol levels (*p* < 0.05). We did not find differences in the prevalence of diabetes mellitus or in specific therapy, including the dose of doxazosin. 

The results from indirect calorimetry between catecholamine phenotypes (Table 6) showed significant differences in VCO2 and RQ, including non-protein RQ. However, no difference was found between phenotypes with respect to the incidence of hypermetabolism. Each hypermetabolic catecholamine phenotype is accompanied by a different substrate metabolism. Under basal conditions, the noradrenergic phenotype preferentially metabolizes carbohydrates, whereas the adrenergic phenotype preferentially metabolizes lipids. The protein oxidation rate between phenotypes remains unchanged. These changes are shown in Figure 1, including the comparison with the control group and percentage of each substrate rate.

Figure 2 shows the relationship between the REE ratio and substrate metabolism rate. In the adrenergic phenotype, there is a positive correlation between percentage of REE ratio and basal lipid rate (R = 0.47, *p* < 0.001), whereas in the noradrenergic phenotype, there is a positive association with basal carbohydrate rate (R = 0.60, *p* < 0.001) and a negative association with lipid rate (R = −0.34, *p* < 0.05). 

## 4. Discussion

Our work focused on the assessment of the incidence of the hypermetabolic state in PPGL patients, its quantification and the influence of the catecholamine phenotype. The results show a high incidence of hypermetabolism in PPGL patients, regardless of catecholamine phenotype. Hypermetabolic patients were older and suffered more from diabetes mellitus and arterial hypertension. Assessment of the basal metabolic rate of each substrate shows that there are differences between the catecholamine phenotypes, with significantly higher lipid and lower carbohydrate consumption in patients with an adrenergic vs. noradrenergic phenotype. These data suggest a different mechanism of action on the energy metabolism of catecholamine phenotypes, as the percentage in REE ratio is closely positively correlated with basal lipid consumption in the adrenergic phenotype, whereas in the noradrenergic phenotype, it is correlated with basal carbohydrate consumption and negatively correlated with lipid rate. 

This work builds on a previous pilot study focused on the presence of hypermetabolism in pheochromocytoma, where a small number of patients did not allow the differences between catecholamine phenotype to be analyzed [3]. Our current data confirm the previous finding that chronic catecholamine overproduction in non-selected PPGL induces a hypermetabolic state. The measured REE was about 16% higher than the predicted value in the whole PPGL group, whereas the measured REE in the control group did not differ from the predicted value. Hypermetabolism was reported in 70% of patients with PPGL, who were more likely to suffer from secondary diabetes and arterial hypertension compared to normometabolic PPGL patients. In the general population, aging is associated with a progressive decline in whole-body REE, whereas in PPGL patients, hypermetabolism has been associated with older age at the time of diagnosis [18].

Our work shows that there are differences between catecholamine phenotype in basal substrate rate metabolism, most likely due to a differing action of each catecholamine. Unfortunately, only a small number of studies have investigated the effect of catecholamines on energy metabolism. Much of our understanding of the regulation of the hypermetabolic state in humans comes from studies simulating the injured state by infusion of hormones into animal models or healthy humans [19,20,21]. However, these studies usually show only the effect of acute or prolonged injury for several hours after administration, but they cannot mimic the long-term effects of high levels of catecholamines on the body. Moreover, the effect of chronic overproduction of catecholamines in PPGL is less predictable due to changes in adrenoceptor profile, affinity, density and sensitivity [12]. Several published studies have shown that elevated plasma adrenaline concentrations induce a hypermetabolic state with increased energy expenditure. Prolonged infusion of adrenaline was found to elevate energy expenditure without producing a concomitant long-term increase in heart rate and blood pressure [19,21]. This increased metabolic rate was associated with an increased oxygen expenditure [19,22,23], probably due to an enhanced glucose oxidation rate [19,24] mediated by β2-adrenoceptor [25]. Some studies have shown that adrenaline directly stimulates lactate release and lipolysis and inhibits insulin-stimulated glucose uptake without affecting amino acid metabolism in the perfused human leg [6]. Furthermore, Glisezinski et al. showed that adrenaline, but not noradrenaline, is a determinant of exercise-induced lipid mobilization in human subcutaneous adipose tissue [7]. The effect of noradrenaline seems to be less important in eliciting metabolic responses [26,27]. In accordance with these studies, we found that the presence of adrenaline, regardless of noradrenaline levels, in PPGL patients leads to more pronounced lipid consumption, whereas pure noradrenaline leads to enhanced carbohydrate oxidation.

Another factor that may be involved in energy metabolism is beige/brown adipose tissue (BeAT/BAT) transformation. Noradrenaline, through beta-3 receptors, induces changes in white adipose tissue that lead to transformation to UCP-1 non-shivering thermogenesis [28]. Glucose and fatty acids (FA) are taken up by the brown and beige adipocytes, and uncoupling protein 1 (UCP1), a marker of BAT, uncouples the oxidative respiration of FA from adenosine triphosphate (ATP) release, leading to the production of heat instead of ATP [29]. Visceral fat “browning” per 18FDG PET/CT and histopathological examination have been shown in patients with PPGL in association with increased energy expenditure and negatively associated with central adiposity [30,31]. However, in our recent work, we found no association between activated beige/brown adipose tissue and hypermetabolism, although we confirmed a correlation between UCP-1 levels and levels of both noradrenaline and its metabolite, normetanefrine [14]. The present study was not designed to consider the presence or influence of BAT/BeAT on energy metabolism in PPGL. However, the two studied groups did not differ in terms of levels of normetanephrine, a metabolite of noradrenaline, and it can be assumed that the possible activation of beige adipose tissue does not explain the differences in metabolized substrates between phenotypes.

Another explanation of metabolic changes between phenotypes is based on genetic mutations and pathogenetic pathways. PPGLs can be classified into three broad clusters, regardless of hereditary or sporadic etiology [2,32]. Cluster 1 is known as the “pseudo-hypoxic” cluster and includes mutations involved in the overexpression of vascular endothelial growth factor (VEGF) and impaired DNA methylation, leading to increased vascularization. Pseudo-hypoxic tumors exhibit a typical noradrenergic secretory profile and are suggestive of mutations in the Krebs-cycle enzymes and hypoxia-signaling pathway PGL-related genes [2]. These mutations cause HIF-2α stabilization, promoting chromaffin/paraganglionic cell tumorigenesis [11,33,34]. In pseudo-hypoxic tumors, HIF-α activation induced the Warburg effect, which is actually a shift in the energy metabolism of tumor cells from oxidative phosphorylation to aerobic glycolysis [11,35]. The alternative pathway of energy generation is less efficient and requires a much greater supply of glucose to sustain the energy needs of tumor cells. Additionally, imaging studies have shown that high 18F-FDG uptake by *SDHx*-related tumors reflects the Warburg effect [36]. Van Berkel et al. showed that the activation of aerobic glycolysis in *SDHx*-related PPGL is associated with increased 18F-FDG accumulation due to accelerated glucose phosphorylation by hexokinases rather than increased expression of glucose transporters [37]. The adrenergic phenotype is often associated with mutations in cluster 2 (kinase-signaling-related tumors), which does not lead to pseudo-hypoxia. Our work was not focused on genetic mutations, and most of the tumors were sporadic, so we can only assess on the basis of the catecholamine phenotype. Tumor somatic mutation was not investigated. However, the group with noradrenergic phenotype had significantly higher rates of basal carbohydrate consumption compared to the adrenergic phenotype group, which correlated with hypermetabolism. This finding indirectly supports the possibility of aerobic glycolysis in the noradrenergic phenotype as the major source of energy and is in line with the reasons why noradrenergic tumors are better visualized in PET/CT with fluorodeoxyglucose [38].

Our study is subject to several limitations. The first limitation is the relatively small sample of patients due to a rare disease. Another important factor that could influence the presented results is pharmacological treatment with alpha-1 and beta-1 competitive blockers. As reported in the literature, this treatment only slightly improves the glucose metabolic disorder, but its improved effect on cardiovascular manifestations is described in [39]. The maximum tolerated doses are low because the escalation of drug doses is complicated by a significant drop in blood pressure or orthostatic hypotension, which is poorly tolerated by patients. Therefore, we believe that our results are interpretable despite treatment with alpha and beta blockers. Another limitation may be the missing data with respect to dietary intake, although all patients were examined during a short hospitalization with a predetermined diet (normal or diabetic diet) and under 12 h fasting conditions. Active smoking is known to increase REE. The control group had a significantly lower number of smokers, which might be a confounding factor with respect to the results from indirect calorimetry. However, the possible slight overestimation compared to controls does not affect the analysis between phenotypes, as the incidence of current smokers did not differ. Finally, although the observed changes in energy metabolism, BMI and body fat indices are likely secondary to the effects of tumor-produced catecholamines, establishing this relationship would require mechanistic studies that are not easily performed in these patient populations.

## 5. Conclusions

In conclusion, our data confirm a high incidence of hypermetabolism in an unselected and consecutive group of patients with PPGL. Assessment of the group according to the catecholamine biochemical phenotype shows a different basal metabolism of main substrates. Whereas the adrenergic phenotype leads to significant lipid consumption, the noradrenergic phenotype leads to more significant carbohydrate consumption. We hope that our work contributes to increased understanding of the action of catecholamines in basal energy metabolism. These are interesting data on energy metabolism, the significance of which remains unclear. Further studies are needed to clarify the consequences of the substrate metabolism of each phenotype.

## Figures and Tables

**Figure 1 biomedicines-10-01980-f001:**
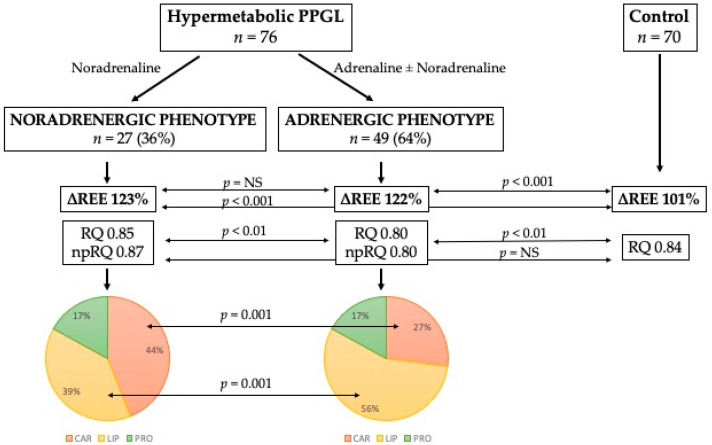
Differences between catecholamine phenotypes in substrate metabolism. Abbreviations: ∆REE, ratio between estimated and measured resting energy expenditure in percentage; RQ, respiratory quotient; np-RQ, non-protein respiratory quotient; CAR, carbohydrates; LIP, lipids; PRO, proteins.

**Figure 2 biomedicines-10-01980-f002:**
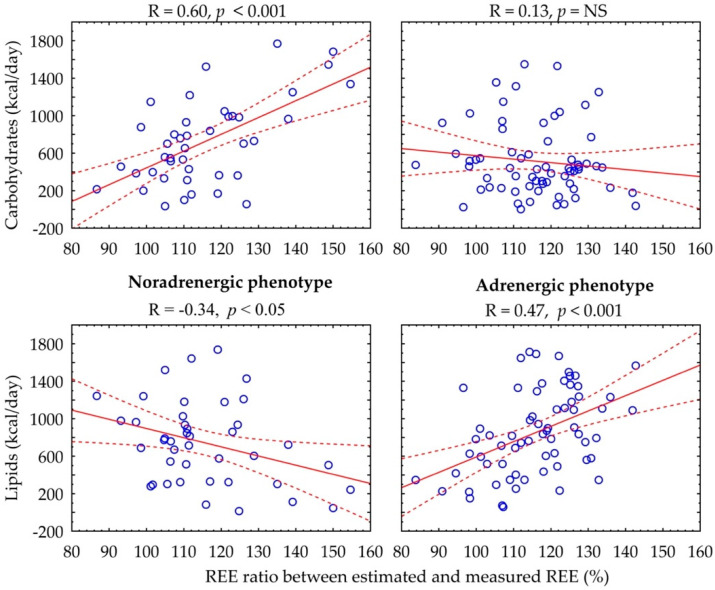
Correlation between carbohydrate and lipid rates and percentage of REE ratio according to catecholamine phenotype.

**Table 1 biomedicines-10-01980-t001:** Basic subject characteristics: PPGL and control groups.

Subjects’ Characteristic	PPGL	Control	*p*
Subjects, *n* (females)	108 (57)	70 (38)	0.97 *
Age, y	51 ± 14	49 ± 15	0.40
Weight, kg	77 ± 20	75 ± 15	0.50
Height, cm	171 ± 10	171 ± 9	0.88
BMI, kg/m^2^	26.3 ± 5.5	25.7 ± 4.2	0.45
Waist, cm	90 ± 16	89 ± 13	0.54
Hip, cm	103 ± 11	103 ± 7	0.99
WHR	0.87 ± 0.10	0.86 ± 0.10	0.41
Body fat percentage, %	32 ± 9	29 ± 8	0.06
Creatinine, umol/L	71 ± 17	74 ± 15	0.27
Type 2 DM, *n* (%)	31 (29)	-	-
FBG, mmol/L	6.0 ± 1.6	5.0 ± 0.6	<0.001
HbA1c, mmol/mol	43 ± 10	36 ± 5	<0.001
Total cholesterol, mmol/L	4.7 ± 1.1	4.8 ± 0.9	0.59
Triglycerides, mmol/L	1.3 ± 0.8	1.6 ± 1.4	0.10
TSH, uIU/L	1.837 ± 1.079	2.212 ± 1.150	0.20
P_Metanephrine, mmol/L	3.0 (0.5; 9.6)	0.2 (0.1; 0.2)	<0.001
Levels above URR	5 (0.9; 18)	0.3 (0.2; 0.4)	<0.001
P_Normetanephrine, mmol/L	9.1 (4.0; 21.2)	0.3 (0.2; 0.5)	<0.001
Levels above URR	12 (5; 27)	0.4 (0.3; 0.6)	<0.001
Current Smoker, *n* (%)	30 (28)	9 (13)	<0.05
Art.hypertension, *n* (%)	73 (68)	-	-
Alpha blockers, *n* (%)	100 (93)	-	-
Dose of Doxazosine, mg	3 (2; 6)	-	-
Beta blockers, *n* (%)	35 (32)	-	-
Statin, *n* (%)	29 (27)	7 (10)	<0.01

Abbreviations: BMI, body mass index; WHR, waist-to-hip ratio; DM, diabetes mellitus; FBG, fasting blood glucose; HbA1c, glycated hemoglobin; TSH, thyroid stimulating hormone; P_, plasma; URR, upper reference range; REE, resting energy expenditure. * Statistical significance is related to the representation of men and women between groups.

**Table 2 biomedicines-10-01980-t002:** Indirect calorimetry results between PPGL and control groups.

Calorimetry Parameters	PPGL	Control	*p*
VO_2_, L/min	0.249 ± 0.051	0.215 ± 0.038	<0.001
VCO_2_, L/min	0.205 ± 0.045	0.181 ± 0.038	<0.001
RQ	0.83 ± 0.08	0.84 ± 0.07	0.28
Measured REE, kcal/day	1734 ± 357	1515 ± 284	<0.001
Predicted REE, kcal/day	1503 ± 301	1493 ± 236	0.80
REE ratio, %	116 ± 13	101 ± 9	<0.001
REE/BSA, kcal/m^2^	918 ± 114	808 ± 92	<0.001
REE/kg, kcal/kg	23 ± 4	20 ± 2	<0.001
REE/FFM, kcal/kg	34 ± 5	29 ± 3	<0.001
Hypermetabolism, *n* (%)	76 (70)	11 (16)	<0.001
Normometabolism, *n* (%)	30 (28)	53 (76)	<0.001
Hypometabolism, *n* (%)	2 (2)	6 (8)	0.08

Abbreviations: VO_2_, oxygen consumption; VCO_2_, carbon dioxide production; RQ, respiratory quotient; REE, resting energy expenditure; BSA, body surface area; FFM, free fat mass.

**Table 3 biomedicines-10-01980-t003:** Basic subject characteristics according to normo- and hypermetabolism.

Subjects’ Characteristics	Normo PPGL	Hyper PPGL	*p*
Subjects, *n* (females)	32 (17)	76 (40)	0.87 *
Age, y	46 ± 14	53 ± 14	0.02
Weight, kg	78 ± 21	77 ± 19	0.71
Height, cm	171 ± 10	170 ± 10	0.54
BMI, kg/m^2^	26.4 ± 5.9	26.3 ± 5.4	0.93
Waist, cm	90 ± 17	91 ± 16	0.75
Hip, cm	105 ± 12	103 ± 10	0.39
WHR	0.85 ± 0.10	0.88 ± 0.09	0.17
Body fat percentage, %	32 ± 9	32 ± 9	0.91
Creatinine, umol/L	76 ± 18	69 ± 16	0.08
Type 2 DM, *n* (%)	5 (16)	26 (34)	0.05
FBG, mmol/L	5.4 ± 0.8	6.3 ± 1.8	<0.01
HbA1c, mmol/mol	40 ± 8	44 ± 11	<0.05
Total cholesterol, mmol/L	4.8 ± 1.0	4.7 ± 1.1	0.71
Triglycerides, mmol/L	1.4 ± 1.1	1.2 ± 0.6	0.49
TSH, uIU/L	2.015 ± 1.032	1.764 ± 1.097	0.22
P_Metanephrine above URR	4.6 (0.8; 14.7)	5.9 (0.9; 21.0)	0.61
P_Normetanephrine above URR	12.8 (4.6; 23.5)	11.2 (5.1; 28.9)	0.79
P_Epinephrine above URR	2.2 (0.5; 4.4)	1.8 (0.5; 6.3)	0.93
P_Norepinephrine above URR	1.5 (1.0; 6.3)	3.7 (1.2; 6.8)	0.17
P_Dopamine above URR	0.5 (0.3; 0.8)	0.5 (0.3; 0.7)	0.50
U_Epinephrine above URR	1.2 (0.2; 4.0)	3.1 (0.4; 15.2)	0.11
U_Norepinephrine above URR	2.1 (0.8; 10.8)	2.9 (1.1; 9.1)	0.84
U_Dopamine above URR	0.7 (0.5; 1.0)	0.8 (0.7; 1.3)	0.74
Current Smoker, *n* (%)	10 (31)	20 (26)	0.77
Art.hypertension, *n* (%)	16 (50)	57 (75)	<0.05
Alpha blockers, *n* (%)	28 (88)	72(95)	0.36
Dose of doxazosin, mg	2 (2; 4)	4 (2; 6)	0.06
Beta blockers, *n* (%)	12 (38)	33 (43)	0.72
Statin, *n* (%)	6 (19)	23 (30)	0.20

Abbreviations: BMI, body mass index; WHR, waist-to-hip ratio; DM, diabetes mellitus; FBG, fasting blood glucose; HbA1c, glycated hemoglobin; TSH, thyroid stimulating hormone; P_, plasma; U_, urine; URR, upper reference range; REE, resting energy expenditure. * Statistical significance is related to the representation of men and women between groups.

**Table 4 biomedicines-10-01980-t004:** Results from indirect calorimetry between normo and hypermetabolism.

Calorimetry Parameters	Normo PPGL	Hyper PPGL	*p*
VO_2_, L/min	0.222 ± 0.044	0.260 ± 0.050	<0.001
VCO_2_, L/min	0.186 ± 0.036	0.213 ± 0.047	<0.01
RQ	0.84 ± 0.08	0.82 ± 0.08	0.15
Measured REE, kcal/day	1552 ± 299	1811 ± 353	<0.001
Predicted REE, kcal/day	1542 ± 300	1487 ± 302	<0.001
REE ratio, %	101 ± 6	122 ± 10	<0.001
REE/BSA, kcal/m^2^	815 ± 72	962 ± 100	<0.001
REE/kg, kcal/kg	20 ± 3	24 ± 3	<0.001
REE/FFM, kcal/kg	30 ± 4	36 ± 4	<0.001
UUN, g/day	13 ± 6	12 ± 4	0.16
np-RQ	0.86 ± 0.11	0.82 ± 0.10	<0.01
Carbohydrates, kcal/d	574 ± 334	605 ± 454	0.72
Lipids, kcal/d	624 ± 375	891 ± 450	<0.01
Proteins, kcal/d	351 ± 152	311 ± 118	0.15

Abbreviations: VO_2_, oxygen consumption; VCO_2_, carbon dioxide production; RQ, respiratory quotient; REE, resting energy expenditure; BSA, body surface area; FFM, free fat mass; UUN, urine urea nitrogen; np-RQ, non-protein respiratory quotient.

**Table 5 biomedicines-10-01980-t005:** Basic characteristics of subject with hypermetabolism according to catecholamine phenotype.

Subjects’ Characteristics	NOR	ADR	*p*
Subjects, *n* (females)	27 (19)	49 (38)	0.68 *
Age, y	47 ± 15	56 ± 11	<0.01
Weight, kg	78 ± 22	76 ± 18	0.55
Height, cm	173 ± 11	169 ± 9	0.07
BMI, kg/m^2^	25.9 ± 5.1	26.5 ± 5.5	0.65
Waist, cm	89 ± 17	92 ± 16	0.59
Hip, cm	102 ± 9	103 ± 11	0.74
WHR	0.87 ± 0.11	0.89 ± 0.09	0.53
Body fat percentage, %	30 ± 7	32 ± 10	0.30
Creatinine, umol/L	71 ± 14	68 ± 17	0.42
Type 2 DM, *n* (%)	7 (26)	17 (35)	0.60
FBG, mmol/L	6.3 ±.2.1	6.3 ± 1.5	0.96
HbA1c, mmol/mol	42 ± 9	46 ± 12	0.22
Total cholesterol, mmol/L	4.3 ± 0.7	4.9 ± 1.2	<0.05
Triglycerides, mmol/L	1.2 ± 0.6	1.3 ± 0.6	0.42
TSH, uIU/L	1.40 ± 0.83	1.97 ± 1.18	<0.01
P_Metanephrine, mmol/L	0.4 (0.2; 0.5)	9.5 (2.7; 14.0)	<0.001
Levels above URR	0.7 (0.3; 0.9)	12.0 (5.0; 25.9)	<0.001
P_Normetanephrine, mmol/L	9.2 (5.6; 26.5)	8.8 (3.2; 10.5)	0.35
Levels above URR	11.7 (7.1; 33.5)	11.1 (4.1; 25.9)	0.81
Current Smoker, *n* (%)	4 (15)	16 (33)	0.16
Alpha blockers, *n* (%)	25 (93)	47 (96)	0.93
Dose of doxazosin, mg	4 (1; 6)	4 (2; 6)	0.93
Beta blockers, *n* (%)	13 (48)	20 (41)	0.71
Statin, *n* (%)	8 (30)	15 (31)	0.86

Abbreviations: BMI, body mass index; WHR, waist-to-hip ratio; DM, diabetes mellitus; FBG, fasting blood glucose; HbA1c, glycated hemoglobin; TSH, thyroid-stimulating hormone; P_, plasma; URR, upper reference range; REE, resting energy expenditure. * Statistical significance is related to the representation of men and women between groups.

**Table 6 biomedicines-10-01980-t006:** Results from indirect calorimetry in hypermetabolism according to catecholamine phenotype.

Calorimetry Parameters	NOR	ADR	*p*
VO_2_, L/min	0.271 ± 0.054	0.254 ± 0.047	0.14
VCO_2_, L/min	0.231 ± 0.049	0.203 ± 0.043	<0.05
RQ	0.85 ± 0.08	0.80 ± 0.08	<0.01
Measured REE, kcal/day	1906 ± 377	1758 ± 332	0.08
Predicted REE, kcal/day	1558 ± 342	1448 ± 273	0.13
REE ratio, %	123 ± 13	122 ± 8	0.49
UUN, g/day	12 ± 5	11 ± 4	0.34
np-RQ	0.87 ± 0.10	0.80 ± 0.10	<0.01
Carbohydrates, kcal/d	832 ± 495	481 ± 381	<0.001
Lipids, kcal/d	734 ± 475	977 ± 415	<0.05
Proteins, kcal/d	329 ± 128	302 ± 113	0.35
Carbohydrates (% of REE)	44 ± 24	27 ± 19	<0.01
Lipids (% of REE)	39 ± 24	56 ± 20	<0.01
Proteins (% of REE)	17 ± 6	17 ± 6	0.97

Abbreviations: VO_2_, oxygen consumption; VCO_2_, carbon dioxide production; RQ, respiratory quotient; REE, resting energy expenditure; BSA, body surface area; FFM, free fat mass; UUN, urine urea nitrogen; np-RQ, non-protein respiratory quotient.

## Data Availability

The data presented in this study are available upon request from the corresponding author. The data are not publicly available due to privacy and ethical principles.

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
