# Peer review of "Hypermetabolism and Substrate Utilization Rates in Pheochromocytoma and Functional Paraganglioma"

_biomedicines, 2022, doi:10.3390/biomedicines10081980_

Round 1

Reviewer 1 Report

Dear authors:

            Kindly presented a clinical research article entitled “Hypermetabolism and basal substrate rates in pheochromocytoma / functional paraganglioma.” The authors aimed to evaluate incidence of a hypermetabolic state and differences in substrate metabolism calculated from calorimetry parameters in patients with pheochromocytoma / functional paraganglioma.

       The significance of this research may be on the prospectively clinical investigation of hypermetabolic state in patients with pheochromocytoma / functional paraganglioma. The study comprised totally clinical data without in vitro experiments and external data comparison.  The experimental arrangement and explanation are a bit redundant and not well-introduced. A few points need to be further clarified. The following are the points essential to be further revised.

1.      In the table 1, the “NS” seems refer to no significance, but the abbreviation was not explained in the table annotation. In addition, as a scientific article, please show up the all the p values, even if the values were taken as not significant.

2.      In the Table 1, the PPGL group had dominantly high percentages with diabetes and hypertension, which might be associated with functional neoplasms. However, compared to the control, predominantly high percentages of current smokers in the patient group might be a confounding factor to the results from indirect calorimetry. The authors should explain and discuss the issue, although it wasn’t cause dominant difference within the patient group.

3.      The basic characteristics and results from indirect calorimetry between normometabolic PPGLs and the control group might be more interesting to be illustrated or discussed, since hypermetabolic PPGLs were reasonably of high values in indirect calorimetry. And the authors might see the some pearls.

Author Response

Dear Reviewer,

Thank you for your comments, suggestions, and the opportunity to improve our manuscript. The main text was modified. Specific answers follow.  

  1. In the table 1, the “NS” seems refer to no significance, but the abbreviation was not explained in the table annotation. In addition, as a scientific article, please show up the all the p values, even if the values were taken as not significant. 

In all tables, the value of statistical significance was expressed numerically.

  1. In the Table 1, the PPGL group had dominantly high percentages with diabetes and hypertension, which might be associated with functional neoplasms. However, compared to the control, predominantly high percentages of current smokers in the patient group might be a confounding factor to the results from indirect calorimetry. The authors should explain and discuss the issue, although it wasn’t cause dominant difference within the patient group.

Active smoking is known to increase REE. The control group had a significantly lower number of smokers, which might be a confounding factor to the results from indirect calorimetry. However, the possible slight overestimation compared to controls does not affect the analysis between phenotypes, where the incidence of current smokers did not differ. We included this information in the limitations of the study.

  1. The basic characteristics and results from indirect calorimetry between normometabolic PPGLs and the control group might be more interesting to be illustrated or discussed, since hypermetabolic PPGLs were reasonably of high values in indirect calorimetry. And the authors might see the some pearls.

Unfortunately, we did not find any significant differences between the normometabolic form of PPGL patients and the control group, except for the expected higher fasting blood glucose and glycated hemoglobin. For that reason, we did not mention in the text of the manuscript. We attach the table.

Normo PPGL

Controls

P

Subjects, n (females)

32 (17)

70 (38)

Age, y

46 ± 14

49 ± 15

0.38

Weight, kg

78 ± 21

75 ± 15

0.49

Height, cm

171 ± 10

171 ± 9

0.41

BMI, kg.m-2

26.4 ± 5.9

25.7 ± 4,2

0.49

Waist, cm

90 ± 17

89 ± 13

0.84

Hip, cm

105 ± 12

103 ± 7

0.46

WHR

0.85 ± 0.10

0.86 ± 0.10

0.73

Body fat percentage, %

32 ± 9

29 ± 8

0.13

Creatinine, umol/l

76 ± 18

74 ± 15

0.72

Type 2 DM, n (%)

5 (16)

-

-

FBG, mmol/L

5.4 ± 0.8

5.0 ± 0.6

<0.01

HbA1c, mmol/mol

40 ± 8

36 ± 5

0. 01

Total cholesterol, mmol/L

4.8 ± 1.0

4.8 ± 0.9

0.81

Triglycerides, mmol/L

1.4 ± 1.1

1.6 ± 1.4

0.51

TSH, uIU/L

2.015 ± 1.032

2.212 ± 1.150

0.96

V O2, L/min

0.222 ± 0.044

0.215 ± 0.038

0.46

V CO2, L/min

0.186 ± 0.036

0.181 ± 0.038

0.54

RQ

0.84 ± 0.08

0.84 ± 0.07

0.79

Measured REE, kcal/day

1552 ± 299

1515 ± 284

0.57

Predicted REE, kcal,day

1542 ± 300

1493 ± 236

0.39

REE ratio, %

101 ± 6

101 ± 9

0.77

REE/BSA, kcal/m2

815 ± 72

808 ± 92

0.75

REE/kg, kcal/kg

20 ± 3

20 ± 2

0.95

REE/FFM, kcal/kg

30 ± 4

29 ± 3

0.09

Reviewer 2 Report

The prospective, single-center clinical study of  Petrák et al. focuses on an interesting topic of clinical importance, i.e. on the hypermetabolic effects of pheochromocytoma and functional paragangliomas (PPGL). They also analyse the effects of catecholamines on intermediary metabolism.

The study is part of a series of research projects focusing on various features of PPGL including cardiovascular and other metabolic effects. The analysis is well-designed and the methods are valid, but the presentation of the results need to be clarified in some instances. The Discussion appropriately summarizes the results, sometimes referring to the results of other projects e.g. involving genetic analysis of PPGL. The references cover the relevant literature, although not all references fit the statements (e.g. reference 11, see point 3. among the comments). However, there are some questions and concerns to be addressed.

Strengths of the study include the complex measurements and complex statistical analyses of energy consumption and intermediary metabolism. The major finding that excessive noradrenerg activity promotes carbohydrate metabolism, whereas excessive adrenalin release with or without increased noradrenalinrelease promotes lipolysis and increased lipid metabolism are also very interesting. 

Limitations include the lack of publication of a pre-study protocol that could decrease the risk of bias. The fact that it appears to be a single-center study also limits the generalizability of the results. Analysis of sex-differences may have also added to the value of the work (in case of sufficient number of participants), since men and women are known to show different characteristics of intermediary metabolism.

Comments:

Title

1.) The title should be more precise: e.g.

Hypermetabolism and substrate utilisation rates in pheochromocytoma and functional paraganglioma

2.) Abstract

Description of the results should be corrected in the Abstract. The Authors describe incorrectly and grammatically questionably:  “In hypermetabolic PPGL, substrate metabolism significant difference among groups was found in VCO2 (P<0.05) and the RQ (P<0.01).” In fact, such differences are found in Table 6, in which differences according to cathecholamine phenotype were compared. [In Table 4, in which VCO2 and RQ of hypermetabolic and normometabolic patients were compared, there was no difference in RQ and the P value for VCO2 was lower than 0.01. And there was also a significant difference in VO2.] Thus, the Authors should decide which results to highlight in the Abstract and they need to do so correctly.

3.) Introduction Line 81

The Authors refer to the hypermetabolic effects of catecholamines giving example of several metabolic pathways “(gluconeogenesis, lipolysis and proteolysis)”. With regard to catecholamines glycogenolysis is a better established effect. The Authors chose reference 11 for this statement which is a Comment on an article titled “Cytokines, the acute-phase response, and resting energy expenditure in cachectic patients with pancreatic cancer.” authored by Falconer, Fearon, Plester, Ross, and Carter. Instead, the following review describes the complex metabolic effects of catecholamines: Hartmann C, Radermacher P, Wepler M, Nußbaum B. Non-Hemodynamic Effects of Catecholamines. Shock. 2017 Oct;48(4):390-400. doi: 10.1097/SHK.0000000000000879.

Materials and Methods

4.) Lines 100-102 The Authors claim that by determining either plasma (N=58) or urinary catecholamines (N=50), they determined also the “own biological action” of the phenotype. How did they decide, whether to determine the catecholamines from the blood or from the urine? Was the urine collected for 24 hours? At what time of the day were the blood samples collected? (Hoemones, e.g. catecholamines show diurnal changes, see e.g. Rao ML, Strebel B, Halaris A, Gross G, Bräunig P, Huber G, Marler M. Circadian rhythm of vital signs, norepinephrine, epinephrine, thyroid hormones, and cortisol in schizophrenia. Psychiatry Res. 1995 Jun 29;57(1):21-39. doi: 10.1016/0165-1781(95)02525-2.)  What do the Authors mean by determining the special (own) biological action of the PPGL phenotype?

Results

5.) Concerning Table 3. The first line showing the number of normometabolic and hypermetabolic patients is shown not to be significantly different, although they appear quite different. Is it a typing mistake or is it really non-significant?

6.) Legends to Table 4 Line 218 it would be easier to understand: „… hypermetabolic PPGLs have higher basal rate of lipid utilisation

7.) Table S1 There are 3 columns. To which comparisons do the P values belong?

Discussion

8.) With regard to the interpretation of the bigger rise in metabolic rate in older individuals, could the lower expectations (baseline) may also contribute to the higher percentage?

9.) Line 330 The sentence seems to be wrong: “… levels of both noradrenaline and its metabolite norepinephrine.” These are synonims. Should not it be normetanephrine, instead of norepinephrine?

Minor remarks

10.) Extensive revision of the manuscript is required, since the language of the manuscript, although comprehensible, contains a lot of grammatical mistakes or doubtful phrasing.

E.g.

Abstract Line 37 „Catecholamines overproduction” should be either „The overproduction of catecholamines” or „Catecholamins’ overproduction”

Abstract Line 41 „… were measure in 108 consequtive PPGL (57 women) and 70 controls (38 women)…” should be „... were measure in 108 consequtive PPGL patients (57 women) and 70 controls (38 women)…”

Abstract Lines 44-45 „Older age, prevalence of diabetes mellitus and arterial hypertension were higher in hypermetabolic PPGL, as compared to normometabolic.” could be „ Older age, prevalence of diabetes mellitus and arterial hypertension showed correlation with hypermetabolic PPGL, as compared to the normometabolic form of the disease.„

Abstract Line 52 „Under basal conditions, noradrenergic preferentially metabolizes carbohydrates, adrenergic phenotype lipids.” should be „Under basal conditions, the noradrenergic type preferentially metabolizes carbohydrates, the adrenergic phenotype lipids.”

Introduction Line 79 „Hypermetabolism, defined by a significant increase by more than 110%...” means that the increase is huge, reaching 210%. An increase by more than 10% will end up producing a 110% resting energy expenditure.

Materials and Methods Line 95 „The study was designed as a prospective study (or project).” or „We carried out a prospective study.”

Discussion Line 275 „Our work focused on…” would be

11.) All abbreviations should be defined when they first appear in the text.

E.g. FDG in Line 99 or PHEO in Line 108 or REE and RQ in Line 135, PGL-related in Line 343

Author Response

Dear Reviewer,

Thank you very much for your comments, suggestions, and the opportunity to improve our manuscript. The main text was modified. Specific answers follow.  

Limitations include the lack of publication of a pre-study protocol that could decrease the risk of bias. The fact that it appears to be a single-center study also limits the generalizability of the results. Analysis of sex-differences may have also added to the value of the work (in case of sufficient number of participants), since men and women are known to show different characteristics of intermediary metabolism. 

This study was part of an already completed grant project “Pheochromocytoma as a model of chronic activation of the stress axis in the pathogenesis of metabolic disorders”, which was realized in years 2016-2019. The pre-study protocol was part of this project, although it was not published in an impact journal. The patient population has the expected sex differences in height, weight, BMI, body fat percentage, and calorimetric parameters. Assessing differences between hypermetabolic and normometabolic men and women with PPGL is complicated by low numbers, which may lead to misinterpretation. A larger group or cooperation with more centers is needed.

Comments:

Title1.) The title should be more precise: e.g. Hypermetabolism and substrate utilisation rates in pheochromocytoma and functional paraganglioma

 Thank you. We modified the title of the article according to your suggestion.

2.) Abstract. Description of the results should be corrected in the Abstract. The Authors describe incorrectly and grammatically questionably:  “In hypermetabolic PPGL, substrate metabolism significant difference among groups was found in VCO2 (P<0.05) and the RQ (P<0.01).” In fact, such differences are found in Table 6, in which differences according to cathecholamine phenotype were compared. [In Table 4, in which VCO2 and RQ of hypermetabolic and normometabolic patients were compared, there was no difference in RQ and the P value for VCO2 was lower than 0.01. And there was also a significant difference in VO2.] Thus, the Authors should decide which results to highlight in the Abstract and they need to do so correctly.

The abstract has been corrected. Particular emphasis was placed on differences between phenotypes.

3.) Introduction Line 81 The Authors refer to the hypermetabolic effects of catecholamines giving example of several metabolic pathways “(gluconeogenesis, lipolysis and proteolysis)”. With regard to catecholamines glycogenolysis is a better established effect. The Authors chose reference 11 for this statement which is a Comment on an article titled “Cytokines, the acute-phase response, and resting energy expenditure in cachectic patients with pancreatic cancer.” authored by Falconer, Fearon, Plester, Ross, and Carter. Instead, the following review describes the complex metabolic effects of catecholamines: Hartmann C, Radermacher P, Wepler M, Nußbaum B. Non-Hemodynamic Effects of Catecholamines. Shock. 2017 Oct;48(4):390-400. doi: 10.1097/SHK.0000000000000879. 

We agree and thank you. Text and citations corrected.

Materials and Methods

4.) Lines 100-102 The Authors claim that by determining either plasma (N=58) or urinary catecholamines (N=50), they determined also the “own biological action” of the phenotype. How did they decide, whether to determine the catecholamines from the blood or from the urine? Was the urine collected for 24 hours? At what time of the day were the blood samples collected? (Hoemones, e.g. catecholamines show diurnal changes, see e.g. Rao ML, Strebel B, Halaris A, Gross G, Bräunig P, Huber G, Marler M. Circadian rhythm of vital signs, norepinephrine, epinephrine, thyroid hormones, and cortisol in schizophrenia. Psychiatry Res. 1995 Jun 29;57(1):21-39. doi: 10.1016/0165-1781(95)02525-2.)  What do the Authors mean by determining the special (own) biological action of the PPGL phenotype?

Our goal was to confirm biological activity of PPGL subjects due to the division into phenotypes. Although all patients had determined plasma metanephrines (biologically inactive metabolites of catecholamines), this does not always mean an overproduction of catecholamines into the bloodstream (silent pheochromocytoma). Therefore, not only the overproduction of metanephrines, but also the confirmed overproduction of catecholamines was sufficient to assess the catecholamine phenotype. Patients without elevation of catecholamines in plasma or urine were excluded (N=5). Urinary catecholamines were collected according to standards. After the previous diet, a 24-hour urine collection was performed, and the urine was acidified. Since this is a less comfortable method for patients, a sample for the determination of plasma catecholamines was also performed in some patients along with samples for metanephrines. Collections were made between 6-7 am. 

Results

5.) Concerning Table 3. The first line showing the number of normometabolic and hypermetabolic patients is shown not to be significantly different, although they appear quite different. Is it a typing mistake or is it really non-significant?

 Statistical analysis refers to the assessment of differences between men and women between groups. It is gender differences that could affect the results, however, all groups had a similar prevalence of men and women. An explanation has been added to the tables.

6.) Legends to Table 4 Line 218 it would be easier to understand: „… hypermetabolic PPGLs have higher basal rate of lipid utilisation

We corrected.

7.) Table S1 There are 3 columns. To which comparisons do the P values belong?

It is an ANOVA statistic, a more detailed post-hoc analysis has been added and the statistical methods section has also been modified.

Discussion

8.) With regard to the interpretation of the bigger rise in metabolic rate in older individuals, could the lower expectations (baseline) may also contribute to the higher percentage?

 We believe that hypermetabolism in older age may be related to a longer unrecognized and undiagnosed disease.

9.) Line 330 The sentence seems to be wrong: “… levels of both noradrenaline and its metabolite norepinephrine.” These are synonims. Should not it be normetanephrine, instead of norepinephrine?

This is a transcription error. Of course, normetanephrine was meant as an inactive metabolite of norepinephrine. We corrected.

Minor remarks

10.) Extensive revision of the manuscript is required, since the language of the manuscript, although comprehensible, contains a lot of grammatical mistakes or doubtful phrasing.

E.g. 

Abstract Line 37 „Catecholamines overproduction” should be either „The overproduction of catecholamines” or „Catecholamins’ overproduction”

Abstract Line 41 „… were measure in 108 consequtive PPGL (57 women) and 70 controls (38 women)…” should be „... were measure in 108 consequtive PPGL patients (57 women) and 70 controls (38 women)…”

Abstract Lines 44-45 „Older age, prevalence of diabetes mellitus and arterial hypertension were higher in hypermetabolic PPGL, as compared to normometabolic.” could be „ Older age, prevalence of diabetes mellitus and arterial hypertension showed correlation with hypermetabolic PPGL, as compared to the normometabolic form of the disease.„

Abstract Line 52 „Under basal conditions, noradrenergic preferentially metabolizes carbohydrates, adrenergic phenotype lipids.” should be „Under basal conditions, the noradrenergic type preferentially metabolizes carbohydrates, the adrenergic phenotype lipids.”

We apologize for the mistakes in the abstract. The text was proofread by a professional English text editor, however, the mistakes were made in an attempt to additionally shorten the abstract to the required 200 words.

Introduction Line 79 „Hypermetabolism, defined by a significant increase by more than 110%...” means that the increase is huge, reaching 210%. An increase by more than 10% will end up producing a 110% resting energy expenditure.

Corrected.

Materials and Methods Line 95 „The study was designed as a prospective study (or project).” or „We carried out a prospective study.”

Corrected.

Discussion Line 275 „Our work focused on…” would be 

Corrected.

11.) All abbreviations should be defined when they first appear in the text. 

E.g. FDG in Line 99 or PHEO in Line 108 or REE and RQ in Line 135, PGL-related in Line 343

Corrected.

Reviewer 3 Report

Dear editor,

I have read with interest the paper entitled " Hypermetabolism and basal substrate rates in pheochromocytoma / functional paraganglioma" (biomedicines-1770470)

This study was conducted including 108 patients with PPGL and 70 patients as control group in order to evaluate incidence of a hypermetabolic state and differences in substrate metabolism calculated from calorimetry parameters in the two groups. The PPGL group was also divided by catecholamine phenotype. The authors found that hypermetabolic state is present in both cathecolamine phenotypes when compared to control group healthy patients. Hypermetabolic PPGL are older and suffer more from diabetes mellitus and arterial hypertension.

The manuscript is very interesting well-structured and contains useful information.

There are some revisions:

-       Considering the several cardiovascular complications in PPGL patients please discuss how the hypermetabolic state should be associated with some of the classic features of PPGL such as blood pressure variability (PMID: 31083609) and PPGL associated cardiomyopathies (PMID: 32709015).

-       Please indicate if there were any differences in patients with metastatic PPGL versus patients with non-metastatic PPGL.

-       Please discuss extensively the association between brown adipose tissue and catecholamines overproduction in PPGL (PMID: 21701596; PMID: 24348550).

Author Response

Dear Reviewer,

Thank you very much for your comments, suggestions, and the opportunity to improve our manuscript. The main text was modified. Specific answers follow.  

There are some revisions:

-       Considering the several cardiovascular complications in PPGL patients please discuss how the hypermetabolic state should be associated with some of the classic features of PPGL such as blood pressure variability (PMID: 31083609) and PPGL associated cardiomyopathies (PMID: 32709015).

Thank you for the topic for further analysis. Hypermetabolism is probably a reflection of the hormonal activity of the tumor, and it is likely that hypermetabolic patients are more at risk of cardiovascular complications. On the other hand, CV complications can arise even in a normometabolic patient after the sudden release of a large amount of catecholamines, e.g. as a result of the use of a secretagogue. We wrote a discussion about CV complications and hypermetabolism in the introduction.

-       Please indicate if there were any differences in patients with metastatic PPGL versus patients with non-metastatic PPGL.

In our group, we had only 3 patients with a metastatic form of PPGL. All three had a noradrenergic biochemical phenotype. A larger patient population would be needed to assess the differences more in detail.

-       Please discuss extensively the association between brown adipose tissue and catecholamines overproduction in PPGL (PMID: 21701596; PMID: 24348550).

The section devoted to brown adipose tissue was slightly expanded and citations were added.